# Tuning the Photophysical Features of Self-Assembling Photoactive Polypeptides for Light-Harvesting

**DOI:** 10.3390/ma12213554

**Published:** 2019-10-30

**Authors:** Maciej Michalik, Mateusz Zbyradowski, Leszek Fiedor

**Affiliations:** 1Faculty of Biochemistry, Biophysics and Biotechnology, Jagiellonian University, Gronostajowa 7, 30-387 Kraków, Poland; maciej.michalik@uj.edu.pl (M.M.); mateusz.zbyradowski@doctoral.uj.edu.pl (M.Z.);; 2Institute of Physics, Jagiellonian University, Łojasiewicza 11, 30-428 Kraków, Poland; 3Ma Chung Research Center for Photosynthetic Pigments, Ma Chung University, Villa Puncak Tidar N-01, Malang 65151, Indonesia

**Keywords:** LH1 antenna, solar energy conversion, spectral tuning, bacteriochlorophyll, carotenoid, reconstitution, pigment exchange, excitation energy transfer, oligomer size, pigment-pigment interactions

## Abstract

The LH1 complex is the major light-harvesting antenna of purple photosynthetic bacteria. Its role is to capture photons, and then store them and transfer the excitation energy to the photosynthetic reaction center. The structure of LH1 is modular and it cooperatively self-assembles from the subunits composed of short transmembrane polypeptides that reversibly bind the photoactive cofactors: bacteriochlorophyll and carotenoid. LH1 assembly, the intra-complex interactions and the light-harvesting features of LH1 can be controlled in micellar media by varying the surfactant concentration and by adding carotenoid and/or a co-solvent. By exploiting this approach, we can manipulate the size of the assembly, the intensity of light absorption, and the energy and lifetime of its first excited singlet state. For instance, via the introduction of Ni-substituted bacteriochlorophyll into LH1, the lifetime of this electronic state of the antenna can be shortened by almost three orders of magnitude. On the other hand, via the exchange of carotenoid, light absorption in the visible range can be tuned. These results show how in a relatively simple self-assembling pigment-polypeptide system a sophisticated functional tuning can be achieved and thus they provide guidelines for the construction of bio-inspired photoactive nanodevices.

## 1. Introduction

The evolution of photosynthesis on Earth began approximately 3.5 billion years ago with the appearance of anaerobic phototrophic bacteria in the oceans, soon (in geological terms) followed by the emergence of oxygenic cyanobacteria, and then by plants [1]. Millions of years of efficient solar energy conversion by photosynthetic autotrophs resulted in a surplus of biomass and molecular oxygen. The accumulation of this very reactive by-product of oxygenic photosynthesis led to the oxidation of the planetary crust and the reductive primary atmosphere being turned into an oxidative one, saturated to a high degree with oxygen. As a result, evolution accelerated and all life forms could flourish throughout the planet [1]. Purple photosynthetic bacteria seem to be a direct descendent of these archaic phototrophic bacteria. Their relatively simple photosynthetic apparatus, composed of a photosynthetic reaction center (RC) surrounded by a modular ring-shaped light-harvesting antenna 1 (LH1), forming together the so-called core complex (RC-LH1), is exceptional in many respects. It seems to be the oldest type of biological photodevice [2] that absorbs photons over a wide spectral range and with a high level of efficiency converts their energy into chemical energy. Bacterial RC is a prototype for RCs present in practically all other types of photosynthetic organisms [2]. In RC, which is the core of this photodevice, solar energy is used, with nearly 100% quantum efficiency, to drive electric charge separation across the photosynthetic membrane, which is then converted into chemically stable intermediates [3]. To achieve such a high level of performance, the photosynthetic apparatus underwent long-lasting evolution and can be regarded as having passed the “longest testing procedure”, during which its design and functioning principles could be perfected.

In most purple bacteria, the RC-LH1 core complex is accompanied by the accessory (or peripheral) LH2 or LH3 antenna complexes, which are structurally similar to LH1 and function to enhance the active cross-section for light-harvesting (LH) [4,5,6]. Although the LH pigment-protein complexes in oxygenic phototrophs are more diverse, the majority of photosynthetic antennae share some key features because of the close similarity of the molecules they bind, carotenoids (Crts) and (bacterio)chlorophylls [7,8], and the role they serve—to fuel the RCs with excitations. Their overall performance is an interplay of numerous factors but there seems to exist no strict structural pattern for the LH antennae, and indeed high functional efficiency in LH can be achieved in several ways [7,9]. The LH1 antenna, directly associated with the RC, is remarkable in terms of its simplistic and modular but highly functional as well as aesthetically pleasing design (see Figure 1) [10,11]. It comprises 15–16 heterodimeric units composed of short hydrophobic α and β polypeptides that bind two types of pigments: bacteriochlorophyll (BChl) and Crt [10,12,13]. The structures of BChl*a* and spirilloxanthin (Spx), the native pigments present in LH1 from *Rhodospirillum (Rsp.) rubrum*, a model organism in photosynthetic research [12,14,15,16], are shown in Figure 1. BChl*a*, which is the major functional (photoactive) and structural component of LH1, is a porphyrin-derived macrocyclic tetrapyrrole that binds Mg^2+^ in the central cavity. The pigment is highly hydrophobic and water-insoluble, owing to the presence of the side residue of phytyl [8]. The other component, Crt, is a long-chain unsaturated isoprenoid that hosts an extensive system of delocalized π-electrons, formed by many conjugated double C–C bonds. The primary role of Crt is the photoprotection of the antenna but it also functions in LH and contributes to structural stabilization of the entire pigment-protein ensemble of LH1 [17,18,19]. In the fully assembled LH1, each α−β heterodimer hosts a dimer of BChl*a* molecules and a Crt molecule. The α−β heterodimers, assembled in a cylinder, serve as a scaffold for a large ensemble of BChl molecules [6,11], whose role is to efficiently capture photons, transiently store the excitation, and funnel the excitation energy to RC, where it is used to drive the photochemical reactions.

The formation of LH1, sometimes referred to as B880, from its monomeric (B780) and dimeric (B820) subunits in micellar medium (*n*-octyl-β-glucopyranoside, β-OG) is schematically presented in Figure 1. The structure of LH1 is exceptional from both the mechanistic and engineering points of view because of its unique self-assembling property that allowed a pigment-protein complex to be completely reconstituted from the isolated components [17,20]. Like other photosynthetic pigment proteins, the polypeptides and BChl are indispensable for assembly, and importantly, information about the spatial organization of the system is encoded in their structures [21]. In effect, the LH1 system in a lipid environment is self-organizing, apparently with no need for external factors [18,20,22]. Our reconstitution approach takes advantage of this feature of LH1 [18,22,23]. In this paper, we show how it can be exploited to allow the essential functional and structural properties of a biological photodevice—the bacterial photosynthetic antenna LH1—to be controlled and tuned. These design principles and traits of photosynthetic antenna LH1 may serve as inspiration for the engineering of biomimetic photodevices for solar energy conversion.

## 2. Materials and Methods

### 2.1. Culturing of Bacteria and Isolation of Chromatophores

The purple photosynthetic bacteria *Rsp. rubrum* S1 and *Rhodobacter (Rdb.) sphaeroides* 2.4.1 were cultivated anaerobically in the Cohen-Bazire medium [24] at 28 °C under continuous illumination with fluorescent white light. The harvested cells were disrupted using a French press (2 × 15,000 psi) and the fraction of chromatophores was separated from the cell debris by centrifugation (50,000 g, 90 min). The chromatophores were stored at −30 °C until use.

### 2.2. Isolation of Pigments

Spheroidene (Sph) was isolated from the cells of *Rdb. sphaeroides* 2.4.1 as described by Fujii et al. [25]. Water and BChl*a* were extracted from the wet cells using cold methanol, and the Crts were then extracted using cold acetone. After fractionation against a saturated aqueous NaCl and *n*-hexane (1:1, v:v), the Crt extract was subjected to two rounds of column chromatography on alumina (ICN, activity II) equilibrated with *n*-hexane. Sph was eluted with 5–10% (v:v) diethyl ether in *n*-hexane, vacuum dried and stored in a mixture of *n*-hexane and tetrahydrofuran (4:1, v:v) at −30 °C.

BChl*a* was isolated using the method described by Omata et al. [26]. A portion of freeze-dried cells of *Rsp. rubrum* S1 was extracted using cold methanol. The extract was dried under vacuum, dissolved in acetone, and loaded on a DEAE-Sepharose CL-6B (Sigma-Aldrich, St. Louis, MO, USA) column pre-equilibrated in acetone. The fraction of BChl*a* was eluted with 20% methanol in acetone (v:v), vacuum dried and stored at −30 °C under Ar. All the solvents were of high performance liquid chromatography (HPLC) grade and purchased from Sigma (Sigma-Aldrich, St. Louis, MO, USA).

### 2.3. Synthesis of Ni-bacteriopheophytin a

Bacteriopheophytin a (BPheo) was prepared by demetalation of BChl*a* in glacial acetic acid. The Ni-substituted BChl*a* (Ni-BPheo) was obtained via transmetalation of a cadmium intermediate (Cd-BPheo) according to Hartwich et al. [27]. Cd-BPheo was synthesized from BPheo with anhydrous cadmium acetate in dimethylformamide. The precursor complex was then transferred to acetone and transmetalated in the presence of anhydrous NiCl_2_ to yield Ni-BPheo. The product was isolated using thin layer chromatography on Silica gel 60 (Merck, Darmstadt, Germany) using a 5% acetone in toluene as an eluent. The purity of the pigments was confirmed spectrophotometrically and using analytical HPLC (see below).

### 2.4. Isolation of Native LH1

The native LH1 antenna from the *Rsp. rubrum* S1 strain was isolated as described previously [23]. Briefly, the bacterial chromatophores were solubilized in 0.45% (*w*/*v*) lauryldimethylamine *N*-oxide (LDAO, Sigma-Aldrich, St. Louis, MO, USA) in Tris-HCl buffer and separated from RC by ultracentrifugation. The pellet of the RC-depleted chromatophores was solubilized in 2.35% (*w*/*v*) β-OG (Carbosynth, Compton, UK) in 20 mM Tris-HCl, pH 7.8, and loaded on a DEAE-cellulose (DE52, Whatman, Maidstone, UK) column pre-equilibrated in 0.8% β-OG (*w*/*v*). The fraction of pure LH1 was eluted with 180–200 mM NaCl in 0.8% β-OG.

### 2.5. Preparation of LH1 Subunits

The LH1 subforms were prepared according to the methods described previously [22,28]. A portion of chromatophores from the *Rsp. rubrum* S1 strain was freeze-dried and depleted of Crts via repeated extraction with benzene, as described previously [23]. The Crt-depleted residue was titrated with aqueous 20% β-OG (*w*/*v*) until B870 was completely dissociated to the B820 form. The insoluble impurities were removed via centrifugation (6000 g, 45 min) and a supernatant that contained crude B820 was loaded on a DEAE-Sepharose FF (Sigma-Aldrich, St. Louis, MO, USA) column equilibrated in a 20 mM Tris-HCl buffer, pH 7.8, containing 1% (*w*/*v*) β-OG. The impurities and free BChl*a* were removed with 10–50 mM NaCl in 1% β-OG and the B820 fraction was eluted with 80 mM NaCl in 1% β-OG. The monomeric B780 form was obtained by titrating B820 with 20% β-OG to a final concentration of ~5% (*w*/*v*), while the oligomeric B870 form was obtained from B820 by lowering the detergent concentration below its critical micelle concentration (CMC), to 0.4% (*w*/*v*).

### 2.6. Pigment Exchange in LH1

Sph was inserted into LH1 according to the method described by Fiedor et al. [23]. Briefly, a powder of Crt-depleted chromatophores was solubilized in 0.3% (v:v) LDAO in 20 mM Tris-HCl buffer to obtain a mixture of B780, B820, and B870. The mixture was titrated with Sph dissolved in acetone until the maximum of the Q_Y_ absorption band was shifted above 880 nm. The reconstituted complex (Sph-LH1) was kept overnight at 4°C in the dark and then purified on a DEAE-cellulose column equilibrated in 0.03% (v:v) LDAO. Pure Sph-LH1 was eluted with 200 mM NaCl in 0.03% LDAO. To insert Ni-BPheo into LH1, a portion of Ni-BPheo dissolved in acetone was added to the B780-B820-B870 mixture before titration with Sph, following a previously published method [29].

To determine the pigment stoichiometry in LH1, the complex was loaded on a small DEAE-cellulose column equilibrated with 0.03% LDAO and thoroughly washed with distilled water to remove the detergent. The pigments were extracted with acetone, dried under vacuum, and the pigment composition of the extract was analyzed using HPLC on a 5 μm LiChrospher Si-60 column (Merck) using a Varian ProStar (Varian, Palo Alto, CA, USA) system equipped with a TIDAS diode array detector (J&M Analytik, Essingen, Germany) for on line measurements of the absorption spectra. The extract was loaded on the column in toluene and eluted with 98.9–95.9% toluene, 0.1% 2-propanol and 1–4% methanol (v:v) as described previously [13].

### 2.7. Spectroscopic Measurements

The electronic absorption and circular dichroism (CD) spectra were recorded in 1 cm quartz cuvettes using a Cary 60 Bio spectrophotometer (Varian, Palo Alto, CA, USA) and a J-815 spectropolarimeter (JASCO, Hachioji, Tokyo, Japan), respectively. The excitation and emission spectra were measured on a Fluoro Max-P fluorometer (Horiba Jobin Yvon, Kyoto, Japan), equipped with a cooled red-sensitive photomultiplier (R2658, Hamamatsu Photonics, Hamamatsu City, Japan) and a 690 nm cut-off filter. The fluorescence emission lifetime was measured using a time-domain Chronos BH fluorometer (ISS, Champaign, IL, USA), with a monochromator set to 900 nm and a H7422P-50 photomultiplier (Hamamatsu Photonics, Hamamatsu City, Japan) as a detector. The excitation pulses of 74 ps duration, output power of 157 mW and 20 MHz repetition rate were generated by a 374 nm or 478 nm laser diode. The emission decay profiles were analyzed using the ISS Vinci software (version 2.0, ISS, Champaign, IL, USA). All measurements were taken at room temperature.

## 3. Results

### 3.1. Isolation of Various Oligomeric Forms of LH1

The subunits of LH1 of various sizes were obtained via detergent-induced dissociation of Crt-free LH1 (B870). The dimeric subunits B820 were isolated from the Crt-depleted chromatophores of wild-type *Rsp. rubrum* by titration with β-OG, until the 820 nm absorption band was saturated, at a final concentration of 3.2% β-OG. The fraction of B820 subunits was isolated using ion-exchange chromatography on DEAE-Sepharose FF in 1% β-OG [22]. By increasing the concentration of β-OG to 5%, the B820 heterodimer was dissociated to the monomeric B780 subunits, comprised of single α and β polypeptides, each coordinating a single BChl*a* molecule [30]. The oligomeric B870 form was obtained via reassociation of B820 in 0.4% β-OG. The association of a B780/B820 mixture in the presence of Crts (in LDAO) yields the B880 form (see below), which can be then purified chromatographically [23].

### 3.2. Electronic Properties

The electronic absorption spectra of the subunits and reassembled B870 complex are shown in Figure 2. The spectrum of B780 resembles that of monomeric BChl*a*, while its dimerization to B820 markedly alters the absorption profile. All the absorption bands are red-shifted but to various degrees (Table 1), the Soret and Q_X_ bands by several nanometers, while the Q_Y_ transition moves from 780 nm to 820 nm, and its intensity substantially increases. The association of B820 into B870 further lowers the energy of the Q_Y_ transition (870 nm) and its intensity remains high, whereas the Soret and Q_X_ bands are less affected. If compared to BChl*a* in solution, the oscillator strength of the BChl*a* Q_Y_ transition increases upon oligomerization, by 50% in B820 and 30% in B870 (Figure 2). Oligomerization of the subunits is accompanied by changes in the CD spectra in the near-IR range (Figure 2). The CD features of B780 are weak but in the case of B820 and B870 very distinct features appear in their spectra, with strong but opposite non-conservative Cotton effects that have broad lobes centered at 792 nm and 830 nm, and at 853 nm and 888 nm, respectively.

### 3.3. LH1 Assembly

The progress of LH1 formation in the presence of Sph as monitored by electronic absorption spectroscopy is shown in Figure 3. The B780 and B820 subunits associate upon the addition of Sph and form B880, i.e., the Crt-binding LH1. The complex formation is complete when the Q_Y_ transition shifts to ~882 nm and the B780 and B820 absorption bands disappear. The absorption maxima of Sph (400–530 nm), from the first stages of reconstitution are red-shifted by ~15 nm with respect to the maxima in acetone, and their positions stay constant during the folding of the antenna (Figure 3). The maxima of Sph absorption are located in the same positions as in the spectra of fully assembled and isolated LH1 [29,32]. The assembly of the subunits can also be induced in the absence of Crt, either by lowering the detergent concentration below its CMC value or by the use of a co-solvent, for instance acetone, and then the B870 complex is formed (not shown) [18].

The Crt-free LH1 antenna from *Rsp. rubrum* was used as a platform to introduce different Crts, also non-native to this species, into the identical BChl*a*-polypeptide context. The absorption spectra in the visible range and the colors of the purified LH1 complexes which were reconstituted with five various Crts, i.e., neurosporene, spheroidene, lycopene, anhydrorhodovibrin and spirilloxanthin (native), are shown in Figure 4. With the increase in the number of double C–C bonds contributing to the conjugated π-electron system, the Crt absorption bands shift to the red and the color of the antenna shifts from green to red (see the photograph in Figure 4). In effect, the Crt absorption bands fill the gap between the BChl*a* absorption bands.

### 3.4. Intra-complex Energy Transfer

The effects of the conjugation length in Crt on the efficiency of intra-complex Crt → BChl energy transfer were examined, comparing the native LH1, which contains Spx, and LH1 reconstituted with Sph (Sph-LH1). The LH capability of these two Crts, incorporated into the same BChl*a*-polypeptide matrix, was determined using a previously described method [33], by comparing the intensities of the fractional absorption and fluorescence excitation spectra normalized to the BChl Q_X_ band (Figure 5). The ratio of spectral integrals between 430 and 560 nm gives an estimate of the efficiency of excitation energy transfer. In the native LH1, the efficiency of intra-complex excitation energy transfer was found to be 40% ± 5%, which agrees with the values previously reported [16,17,34]. In Sph-LH1 it is markedly higher, between 85% and 95% which is significantly different from 58–75% previously found in the LH1 from *Rdb. sphaeroides* which contained the same Crt, either native or reconstituted [17,34,35].

### 3.5. Excited-State Properties

To assess how the properties of the first excited singlet state of polypeptide-bound BChl*a* change depending on the oligomerization state of the pigment, the fluorescence lifetime (τ_fl_) of LH1 and its subunits was determined using picosecond time-resolved fluorometry. The values of τ_fl_ are similar to the one reported previously [30,36] and very strongly depend on the degree of LH1 subunit oligomerization. The emission of fluorescence from the monomeric pigment is relatively long-lived (Table 1), with τ_fl_ ranging from 2.3 to 2.9 ns. In the oligomeric forms τ_fl_ drastically decreases to a sub-ns range, between 600 and 800 ps.

The LH1 reconstitution technique facilitates the insertion of modified BChls into the antenna complex. This approach was used to partially replace the native pigment with its Ni-substituted analog, Ni-BPheo. This substitution does not lead to significant changes in the electronic absorption and CD spectra of LH1 (not shown), whereas its S_1_ state features change drastically, as illustrated in Figure 6. The emission is quenched completely when the content of Ni-BPheo in LH1 reaches 15% regarding the native BChl*a* (Figure 6A). At the same time, the emission decay profile becomes bi-exponential, which contrasts a mono-exponential decay in the non-substituted LH1 (Figure 6B). The single emission component detected in the latter case corresponds to a τ_fl_ value of 790 ps. The same component (τ_fl_ ~800 ps) is found in the emission from LH1 containing Ni-BPheo, in addition to a very short-lived emission, with τ_fl_ < 1 ps (below the time resolution of the apparatus). The contribution of the long-lived component decreases with the growing content of Ni-BPheo incorporated in LH1 (not shown).

## 4. Discussion

### 4.1. Tuning of the Oligomeric State

The equilibrium between the LH1 subunits B780 ↔ B820 ↔ B870/880 can be pushed towards oligomerization by enhancing the polypeptide-cofactor and polypeptide-polypeptide interactions, which can be achieved in several ways. At high concentrations of surfactant (5% β−OG) the antenna is completely dissociated into the monomeric B780 subunits. Upon lowering of the surfactant concentration, the subunits dimerize to B820, which further associate to B870. The subunit oligomeric state is reflected in the characteristic position of the Q_Y_ absorption band and shape of the CD spectrum, which facilitates a convenient monitoring of the oligomerization stages (Figure 3). The assembly of LH1 is accompanied by a gradual red-shift of the Q_Y_ transition of BChl*a* from 780 nm to 880 nm, while in the CD spectrum a strong Cotton effect appears in the corresponding spectral regions, with the sign inverted when going from B820 to B870. Depending on the oligomer size, the oscillator strength of the Q_Y_ transition of BChl*a* increases by 50% (B820) and 30% (B870), which is indicative of an increase in the dipole strength of the lowest energy transition (see below).

The introduction of Crt and organic co-solvent into the reconstitution mixture also markedly affects the equilibrium between B780, B820, and B870/880. Crt triggers the assembly of B880 even at a very low Crt-to-subunit ratio, which indicates a high affinity of Crt to its binding sites and cooperativity of Crt-subunit association and cooperative folding of LH1 [23]. The red-shift of the absorption bands of LH1-bound Crt, as observed from the onset of reconstitution, also speaks for strong interactions between Crt and the α and β polypeptides. The driving force for complex assembly is mostly provided by π–π stacking between Crt and aromatic residues at the core of the helical transmembrane regions of the polypeptides [37], as well as via hydrophobic interactions [38]. Acetone, the co-solvent used during LH1 reconstitution, affects the properties of both the medium, by lowering its permittivity, and surfactant, by increasing the CMC value. In effect, electrostatic interactions between the interfaces of the transmembrane helical fragments of the α/β polypeptides are greatly intensified and the thermodynamics of LH1 assembly in a micellar medium is under the strong control of co-solvent and the Crt cofactor [18,38]. A similar stabilizing effect of Crts was found in LH2 [33]. In the absence of these external factors, the interaction energy (ΔH°) between the LH1 polypeptides equals −580 kJ/mol. In the presence of acetone, ΔH° increases to −1160 kJ/mol, and then to as much as −1900 kJ/mol in the presence of Crt. Intriguingly, despite these stabilization effects of Crt and co-solvent, and the increase in the inter-subunit interaction energy, the driving force for the association remains almost constant, apparently due to efficient enthalpy-entropy compensation in the system.

### 4.2. Tuning the Light-Harvesting and Excited-State Features

The spectral range and the intensity of light absorption are among the most important aspects of LH using photosynthetic antenna complexes and in LH1 they can be controlled by varying the aggregation state of the subunits. Thus, the key photophysical features of BChl*a*, its major chromophore, are tuned in such a way that the pigment S_1_ state energy is lowered, which shifts the light absorption from the far-red edge of the visible range to the near-IR region. At the same time, due to hyperchromism, the cross-section for light absorption in this spectral region increases considerably. Two mechanisms seem to be responsible for such a sophisticated level of tuning available to the purple photosynthetic bacteria. The S_1_ energy is lowered due to excitonic interactions between BChl*a* molecules, which in turn enables a certain intensity borrowing to the S_0_ → S_1_ transition from the higher energy transitions (see below) [36,39].

Light absorption in the blue-green spectral region can be tuned as well, via modifications of the Crt cofactor. As illustrated in Figure 4, this is achieved by varying *n*—the size of the delocalized π-electron system of Crt. In effect, the gap between the Soret and Q_X_ transitions in BChl*a* (see the absorption spectra in Figure 4 and Figure 5), can be filled up with the absorption bands of Crts and this improves the capturing of photons in the visible range. However, this improvement comes at a certain cost, specifically the efficiency of the intra-complex Crt → BChl energy transfer is somewhat compromised. From almost 90%, when the antenna binds Sph (*n* = 9), it drops to ~30% in either native or reconstituted LH1 that contains Spx (*n* = 13). The most plausible reason for this dependence is the shortening of the lifetime of the S_2_ donor state of Crt as *n* is increasing [16,40]. Efficient and fast intra-complex Crt → BChl energy transfer takes place only if Crt is incorporated into properly folded LH1 [17] and hence its efficiency, being strongly dependent on the mutual orientation and inter-pigment distances [33], can be regarded as an indicator that the assembled antenna complex is functioning properly. A very high efficiency of energy transfer in Sph-LH1, reaching 90%, confirms that a fully functional LH1 complex incorporating the non-native Crts has been reconstituted [16,23]. In this context, it should be noted that in the native photosynthetic antennae neither LH nor structural stabilization but the photoprotection seems to be the most important role of Crts [19].

The red-shift of the Q_Y_ band and the characteristic shape of the CD spectra of B820 and B870 show exciton splitting of the energy levels of BChl*a* in LH1, which is due to excitonic coupling in the ensembles of chromophores [41]. Such an interaction is observed when transition dipole moments of participating pigments are physically very close (< 10 Å), and their mutual orientation is in head-to-tail geometry [42]. This implies a specific arrangement of BChls in B820, which is achieved through association between the α/β polypeptides, induced via ion-pairing and hydrogen-bonding of complementary N-terminal domains in each polypeptide [43]. π–π stacking between paired BChls and pigment-polypeptide and polypeptide-polypeptide hydrophobic interactions, which involve the phytyl moiety of BChl and α-helical core of the polypeptide, also promote dimerization/oligomerization [18,43,44]. The assembly of B820 subunits into a B870 complex brings about a further red-shift of the Q_Y_ band and characteristic CD features in the near-IR region (Figure 2), which suggests that excitonic coupling is enhanced, involving both intra- and inter-dimer interactions that engage a larger number of chromophores. Such spectral changes reflect the formation of a supercomplex, i.e., a large array of electronically coupled BChl*a* molecules. The electronic excitations in such systems are delocalized over several pigment molecules [45]. Another indication of exciton delocalization is an increased radiative rate in a molecular ensemble [36], the so-called superradiance, which is manifested in the shortening of the S_1_ state lifetime, as reflected in the fluorescence lifetimes of the oligomeric forms of BChl*a* (see Table 1).

The interactions between pigment molecules and the α and β polypeptides in the assembled antenna are responsible for the tuning of excitonic interactions, which is achieved by maintaining a specific spatial arrangement of chromophores. Due to such tuning of BChl by the polypeptides in LH1, the quenching effects that would be expected to occur at a high local concentration of the pigment are also diminished [6,46]. The same design principle is exploited in the peripheral LH2 complexes of purple bacteria for collecting solar energy following the excitation sink principle, whereas the excitonic coupling of chromophores in LH1 allow excitations to concentrate near the RCs [6].

The chemical engineering of (bacterio)chlorophylls opens further possibilities for the tuning and modifications of the photosynthetic antennae. A very useful seems to be the substitution of central Mg^2+^ ion with divalent transient metal ions, such as Cu^2+^, Ni^2+^, Co^2+^, Pd^2+^ or Pt^2+^, owing to their specific mode of bonding to the core nitrogen atoms. Due to the unfilled d shell, such cations form strong mixed coordination-covalent bonds in the central cavity of (bacterio)chlorophylls [47,48]. This brings about novel photophysical features in the metallosubstituted pigments [47,49], while their relevant chemical properties, such as axial ligand coordination, remain unchanged. This facilitates their interactions with pigment-binding polypeptides and enables pigment exchange in LH1. Owing to the exceptional mode of the central metal binding, the internal conversion and/or intersystem crossing in the metallocomplexes are strongly affected [50,51,52]. In the extreme case, radiationless decay of the excited singlet state of Ni-BPheo takes place in less than 100 fs, during which the electronic excitation is converted into heat with 100% efficiency [47], which may be particularly advantageous. Ni-BPheo readily replaces BChl*a* in LH1 and, with an extremely short-lived S_1_ state, it can be used as an ultrafast excitation trap to probe energy migration pathways and exciton delocalization in both LH1 [13,53] and RC complexes [54]. The transient femtosecond absorption measurements in the LH1 complexes that contain Ni-BPheo showed that exciton delocalization over BChl ensemble in LH1 is large and involves at least 10 BChl*a* molecules. Furthermore, a single Ni-BPheo molecule per 20 BChl*a* molecules suffices to entirely quench LH1 fluorescence [13]. A very similar quenching effect is seen in LH1 from *Rsp. rubrum*, reconstituted with varying amounts of Ni-BPheo (see Figure 6). The incorporation of the Ni-analog of BChl*a* as the excitation trap drastically shortens the fluorescence emission lifetime of LH1 which gives rise to another level of control of the antenna excited states.

## 5. Conclusions

The self-organizing polypeptide-cofactor system of the LH1 antenna from purple photosynthetic bacteria is a prominent example of a tunable biological photodevice. The tuning in LH1 is multi-level and concerns almost every aspect of the system; from its size and subunit oligomerization state, through the thermodynamics of its assembly, to its photophysics and LH features. There are several ways to control the reversible assembly of LH1 in lipid/micellar media, by adding a Crt cofactor and/or a co-solvent, or varying the temperature or the concentration of surfactant. The changes in the LH1 subunit oligomerization state are paralleled by sophisticated tuning of the photophysical features of the major chromophore, BChl*a*, to optimize the LH and excitation energy transfer by the antenna. The Crt exchange approach enables the LH capability of LH1 to be tuned to take better advantage of photons available in the visible spectral region and to improve the efficiency of the intra-complex excitation energy transfer, in addition to photoprotection of the system. Another level of LH1 tuning is feasible via chemical modifications and exchange of the cofactors.

The bacterial photosynthetic antenna LH1 is an exceptionally versatile model system in which the principles of tuning the key functional features can be demonstrated as well as investigated in detail. The system is self-organizing in the lipid environment and the information required for its assembly is encoded in the structures of its components. Very significantly, LH1 design facilitates sophisticated functional tuning of its chromophores, resulting in highly efficient LH over a broad spectral range, the excitation storage and its concentration towards RC, and the photoprotection, which is crucial in a system with a continuous energy flow through. Obviously, however, the biosynthetic costs of such a photoactive biological device are very high. Its assembly in the bacterial photosynthetic membranes requires a concerted cooperation between its components—three different types of biochemical entities, each being the product of a different biosynthetic pathway. Despite that, LH1 provides us with numerous and challenging design guidelines for the construction of man-made photodevices for efficient solar energy harvesting and conversion.

## Figures and Tables

**Figure 1 materials-12-03554-f001:**
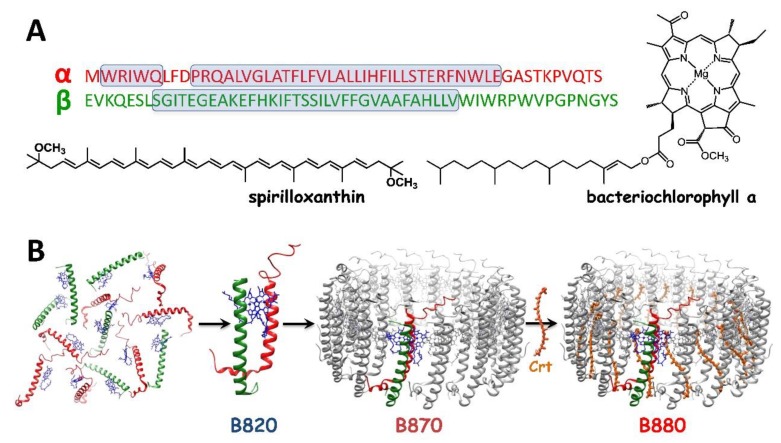
(**A**) The amino-acid sequences and structural formulae of the components of the LH1 antenna from the purple photosynthetic bacterium *Rhodospirillum rubrum*. (**B**) A scheme showing the assembly of the carotenoidless (B870) and carotenoid-binding LH1 complex (B880) from the monomeric (B780) and dimeric (B820) subunits, and a carotenoid (Crt), in a micellar environment. The sizes of the complexes are not to scale. The structure of LH1 is taken from the Protein Data Bank (entry 5Y5S).

**Figure 2 materials-12-03554-f002:**
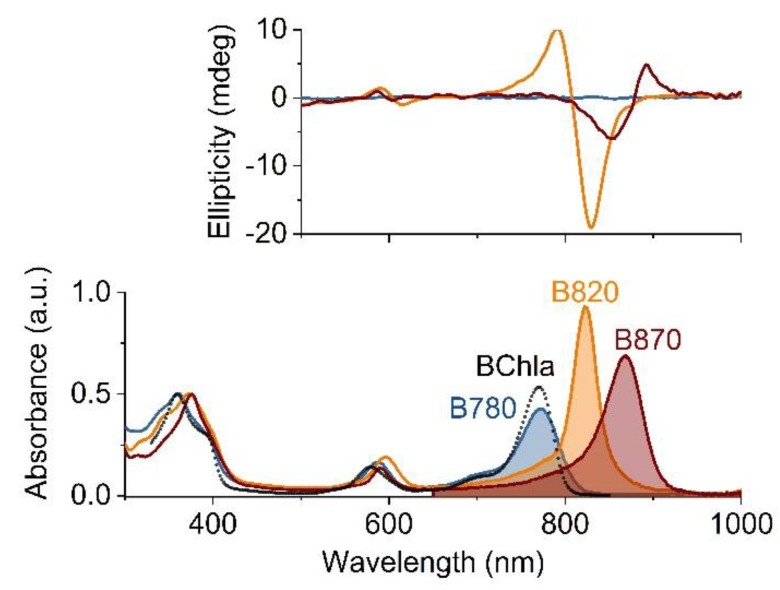
The electronic absorption (lower panel) and circular dichroism (upper panel) spectra of the B780 (blue), B820 (orange) and B870 (dark pink) oligomeric forms of LH1 in 5%, 1% and 0.4% aqueous β-OG, respectively, recorded at room temperature. The absorption spectra were normalized to the maximum of the Soret band whereas the circular dichroism spectra to the absorption intensity of the Q_Y_ maximum. For comparison, also the absorption spectrum of monomeric bacteriochlorophyll a (BChl*a*) in acetone is shown (black dotted line). The shaded areas denote the integrated intensities of the Q_Y_ transitions.

**Figure 3 materials-12-03554-f003:**
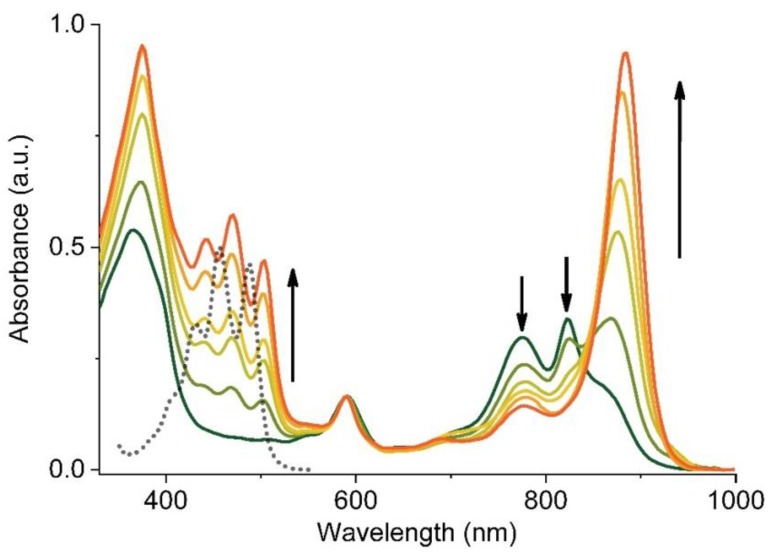
Spectral changes recorded during a Sph-induced reassembly of LH1 from its B780 and B820 subunits in micellar environment of LDAO. The spectra were normalized to the Q_X_ band. The arrows indicate the direction of the changes in the spectra. For comparison, the absorption spectrum of Sph in acetone is also shown (black dotted line).

**Figure 4 materials-12-03554-f004:**
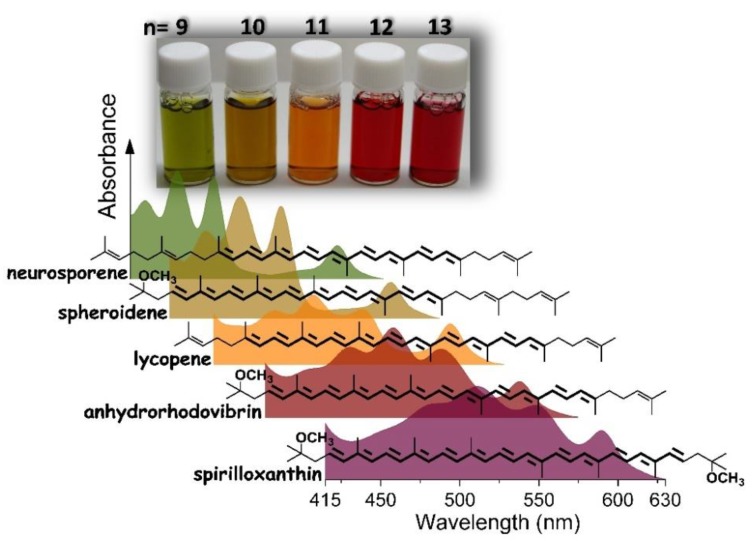
The electronic absorption spectra and coloration (the photograph in the upper part) of LH1 reconstituted with five different carotenoids with the length of their system of conjugated double C–C bonds (*n*) varying from 9 to 13. The structural formulae of the carotenoids are shown above the corresponding absorption spectra.

**Figure 5 materials-12-03554-f005:**
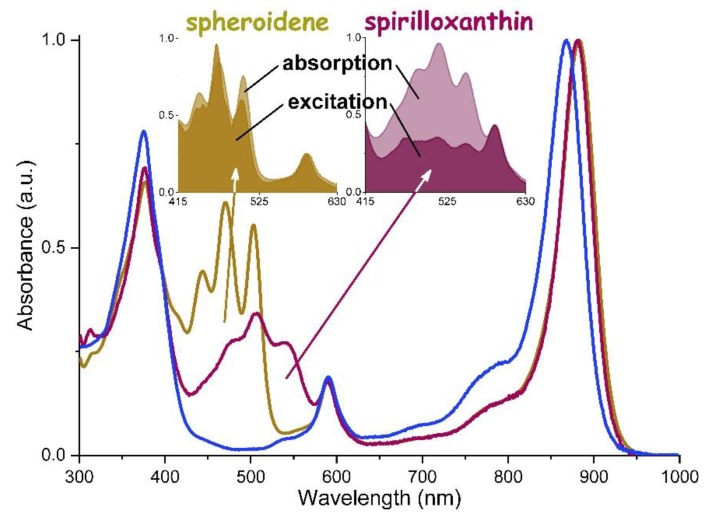
The electronic absorption spectra of B870 (blue line) and LH1 reconstituted either with spheroidene (dark yellow line) or spirilloxanthin (purple line). In the insert, the overlapped absorption and fluorescence excitation spectra of the two complexes are shown. The spectra were normalized to the Q_Y_ band.

**Figure 6 materials-12-03554-f006:**
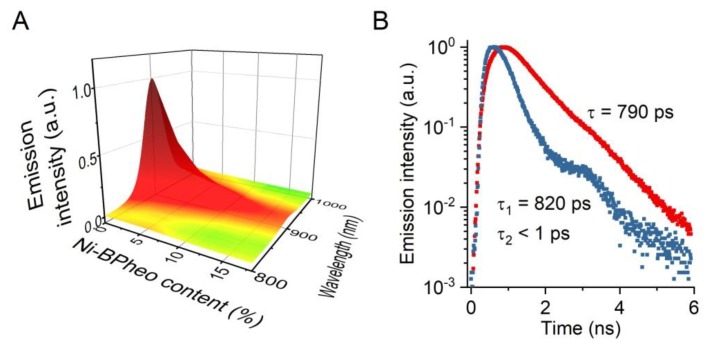
(**A**) The steady-state emission spectra of LH1 substituted with varying amounts of Ni-BPheo. (**B**) Fluorescence decay profiles of LH1 substituted with 0% (red trace) and 18% (blue trace) of Ni-BPheo, with the fitted emission lifetime components. The traces were normalized to the maximum intensity. The optical density of the samples at the excitation wavelength (478 nm) was 0.1 cm^−1^.

**Table 1 materials-12-03554-t001:** The absorption and emission maxima, and the first singlet excited-state lifetime (τ_fl_) of BChl*a* in acetone, the monomeric (B780), dimeric (B820) and oligomeric (B870) subunits of LH1, the LH1 reconstituted with Sph (Sph-LH1) and the native LH1. The excitation wavelength was 374 or 478 nm.

	Major Absorption Maxima (nm, cm^−1^)			Fluorescence Maximum (nm)	τ_fl_ (ns)
	Soret	Q_X_	Q_Y_		
BChl*a*	357, *28011*	579, *17271*	770, *12987*	775	2.9 *^a^*
B780	361, *27701*	586, *17065*	780, *12821*	791	2.30 ± 0.003
B820	370, *27027*	595, *16807*	821, *12180*	833	0.77 ± 0.02
B870	375, *26667*	590, *16950*	869, *11507*	887	0.61 ± 0.003
Sph-LH1	377, *26525*	591, *16920*	883, *11325*	904	0.79 ± 0.001
native LH1	377, *26525*	590, *16950*	881, *11351*	902	0.85 ± 0.02

*^a^*: value taken from Karcz et al. 2014 [31].

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
