# Peer review of "Tuning the Photophysical Features of Self-Assembling Photoactive Polypeptides for Light-Harvesting"

_materials, 2019, doi:10.3390/ma12213554_

Round 1

Reviewer 1 Report

The manuscript titled “Tuning the photophysical features of self-assembling photoactive polypeptides for light harvesting reports on the photophysical properties and controlled assembly of bacterial photosynthetic antenna.

The work was carefully planned and conducted, the results are properly presented, and the manuscript is quite well written. Overall this work can be accepted without revision.

Author Response

We highly appreciate the comments.

Reviewer 2 Report

Authors report on manipulation of LH1 light harvesting antenna (of the purple photosynthetic bacteria). Overall the work appears to be thorough and well written. Following areas might require a minor revision. First, Paragraphs 1 and 2 in the introduction do not seem to be well connected. The second paragraph poses too many jargons and abbreviations that are not well registered by the end of first paragraph. Authors are advised to shorten the second paragraph and make a clear link to their experiment in the third paragraph. Second, the electronic absorption spectrum in Figure 2 does not show BChla, but the authors still make a reference in their main text while stating that the ‘spectrum of B780 resembles that of monomeric BChla‘. Authors are advised to either include the comparative spectra or consider rephrasing their statement listed in the main text. Third, the conclusion section seems very long - it needs to be rewritten to state their key findings and provide a future outlook. Lastly, the line-spacings are off in many places, e.g., Page 12, first vs second paragraph.

Author Response

We are very thankful for all the positive as well as critical comments, and we have complied to most of them. The changes are marked blue-red in the revised manuscript and the major changes are the following:

In Fig. 2 the absorption spectrum of bacteriochlorophyll a added and referred to in the text, as suggested the definition of the core complex added to make a connection between paragraph 1 and 2 (lines 43-44), as suggested The Introduction is now shortened - details of the RC-LH1 complex removed (lines 85-92), as suggested The definitions of abbreviations were carefully re-checked Concerning the line spacing, the formatting of the text is done automatically by the template

Reviewer 3 Report

The topic under discussion is interesting. The authors present a light capture antenna made with purple LH1. The results show its good operation and the process of multilevel tuning.

In order to improve the document, the following changes are recommended:

The abstract should better reflect the most relevant contributions of results and conclusions. The document requires a more up-to-date in-depth review of its references, only 3 of 57 references are from 2017. For example, with the keyword "LH1 antenna" in Scopus there are 247 documents, 33 from 2015 to 2019. I reviewed a document with an anti-plagiarism program (Turnitin) and it gives an 17% plagiarism, It does not signify plagiarism, as there are preconceived phrases that are difficult to eliminate, but one should try to reduce the percentage whenever possible in order to make evident the originality of the document. Please check.

Author Response

We are very thankful for the positive as well as critical comments. The major changes in the text are as follows:

As recommended, wherever possible, older references have been replaced with newer ones, positions 2, 3, 19, 33 and 42, and a reference to the reconstitution of bacteriorhodopsin is now added - position 21 Concerning the Abstract, there is a 200 word limit and we think all the major conclusions from our work are included within this limited space We have checked the text for "plagiarisms" and we agree that most of the indications made by the program are due to specific terminology and language used in the field, but it is difficult to avoid.

Reviewer 4 Report

The work submitted by Michalik and coworkers summarises the main results on the tuning of photophysical features of LH1 complex for light harversting.

The manuscript is in general well written and organised. The topic is of soundness and fits to the scope of this journal. Tables and figures support the explanations in the text. 

However, some minor issues must be addressed by the authors before its publication.

Comments have been embedded through the text in order to help the authors.

Author Response

We are very thankful for the positive as well as the critical comments. According to the suggestions, the following changes have been made:

A reference to the reconstitution of bacteriorhodopsin is added - position 21 The term „cofactor” is now replaced by "pigment" or "component" All the details of the experimental procedures are given in the proper references, as indicated in the text, and in order to avoid unnecessary repetitions we provide only brief descriptions of the procedures.